# Resolving cell state in iPSC-derived human neural samples with multiplexed fluorescence imaging

Martin L. Tomov[1,2,3], Alison O'Neil [1,2], Hamdah S. Abbasi[4], Beth A. Cimini[4], Anne E. Carpenter [4], Lee L. Rubin [1,2,5✉] & Mark Bathe [3,5✉]

Human induced pluripotent stem cell-derived (iPSC) neural cultures offer clinically relevant models of human diseases, including Amyotrophic Lateral Sclerosis, Alzheimer's, and Autism Spectrum Disorder. In situ characterization of the spatial-temporal evolution of cell state in 3D culture and subsequent 2D dissociated culture models based on protein expression levels and localizations is essential to understanding neural cell differentiation, disease state phenotypes, and sample-to-sample variability. Here, we apply PRobe-based Imaging for Sequential Multiplexing (PRISM) to facilitate multiplexed imaging with facile, rapid exchange of imaging probes to analyze iPSC-derived cortical and motor neuron cultures that are relevant to psychiatric and neurodegenerative disease models, using over ten protein targets. Our approach permits analysis of cell differentiation, cell composition, and functional marker expression in complex stem-cell derived neural cultures. Furthermore, our approach is amenable to automation, offering in principle the ability to scale-up to dozens of protein targets and samples.

---

[1] Stanley Center for Psychiatric Research, Broad Institute of MIT and Harvard, Cambridge, MA, USA. [2] Department of Stem Cell and Regenerative Biology, Harvard University, Cambridge, MA, USA. [3] Department of Biological Engineering, MIT, Cambridge, MA, USA. [4] Imaging Platform at Broad Institute of MIT and Harvard, Cambridge, MA, USA. [5]These authors jointly supervised this work: Lee L. Rubin, Mark Bathe. ✉email: lee_rubin@harvard.edu; mark.bathe@mit.edu

Induced pluripotent stem cell (iPSC)-derived cultures are increasingly becoming the principal source of patient-specific and disease-specific cellular material for in vitro disease modeling. iPSC-derived cortical and motor neuron cultures have successfully been used to model neurodevelopmental conditions including autism spectrum disorder (ASD; cortical neurons)[1–5] and neurodegenerative conditions including spinal muscular atrophy and amyotrophic lateral sclerosis (SMA, ALS; motor neurons)[6–9]. Such iPSC-based models are attractive because they can generate the large numbers of neural cells needed for drug screening. They can also recapitulate aspects of cortical and motor neuronal synaptic networks, which allow for functional models of neurodevelopmental and neurodegenerative conditions to be developed in vitro. Further, new iPSC 3D organoid culture methods can generate the various cell types and architecture of a specific tissue[10,11]. However, phenotypic characterization of these cultures is challenging due to their complexity, heterogeneity, and variability; this provides a clear opportunity for improved high content analysis techniques, especially those that can produce multidimensional readouts from the same culture.

Fluorescence-based antibody staining of target markers is one approach to characterizing in situ stem-cell derived neural cultures, but is limited by the conventional spectral limit of fluorophores to imaging four targets simultaneously. Several techniques[12–14] have been reported that bypasss this imaging limitation. Here we characterize one such technique to overcome this limitation that uses DNA-conjugated antibodies to fluorescently image more than ten individual markers in the same sample[15,16]. While the technique still only images up to four markers at a time, the reversible and gentle washing steps that allow for rapid fluorophore-conjugated oligo cycling can produce multidimensional imaging data that can characterize complex cell cultures with many more markers. The procedure we describe here, called PRobe-based Imaging for Sequential Multiplexing (PRISM), eliminates the need for antibody stripping or removal for multiplexing, a limiting factor in traditional immunofluorescence/immunocytochemistry (IF/ICC)-based multiplexed imaging strategies[17,18], thereby allowing characterization using a greater number of molecular targets. Briefly, PRISM antibodies use short, orthogonal oligonucleotide sequences that are complementary to either a fluorophore-conjugated locked nucleic acid (LNA) or DNA strand that reversibly hybridizes to produce a fluorescent readout, analogous to traditional IF/ICC imaging[16,19,20] and DNA-PAINT/EXCHANGE-PAINT[19]. As LNA can be somewhat challenging to generate in large quantities, we also explored the ability of DNA oligos to supplement PRISM imaging, while maintaining high signal-to-noise ratios and marker specificity. We were able to completely reduce non-specific binding of DNA-only PRISM oligos to endogenous nucleic acid sequences, which allows for a more straighforward design and implementation of PRISM-based imaging techniques. Further, compared to standard antibody stripping procedures[21–23], PRISM offers non-destructive imaging probe exchange, cycling fluorescent imaging strands within several minutes permitting multiple rounds of imaging data acquisition[24–26] from the same culture. These features allow for the use of large panels of markers, consequently providing higher content datasets. Generation of oligo pairs is relatively straightforward due to the use of commercially available thiolated and fluorescently labeled oligo strands, which can be readily conjugated to a wide variety of commercially available antibodies[16,19,20].

Here, we apply PRISM to stem cell-derived cortical and motor neuron cultures to characterize cell identity and population composition based on detection of structural, synaptic, and transcription factor markers (Fig. 1). Identification of multiple cell states within the same iPSC-derived 2D or 3D sample helps spatially map the temporal evolution and heterogeneity of targets in human samples, in addition to characterizing structural and synaptic features pertinent to human disease phenotypes[27–29]. We also introduce a CellProfiler/FIJI computational pipeline to analyze imaging data regarding cell state/phenotype in an automated, quantitative, unbiased manner[28,29]. 2D cultures can be used for high-throughput characterization of stem cell-derived neurons, which lends itself well to automation assays. We hypothesize that this technique could be adapted to 3D organoid sections, and as these retain the higher order structure of cellular organization, especially in neural cultures, this may be critical for disease modeling in vitro. Because PRISM is a direct analog to IF/ICC imaging, our approach complements assays in model culture systems currently used to evaluate stem cell differentiation, compound and drug candidate screening, tissue engineering, and potentially in vitro disease diagnostics development.

## Results

We built 12 antibodies into a PRISM panel to characterize the cellular composition of iPSC-derived CN and MN cultures. Specificity of target imaging was confirmed using traditional IF and compared to respective PRISM markers. We used established stem cell-derived astrocyte, CN, and MN cultures to screen a large panel of neural markers (Supplementary Table 1)—before narrowing down to a panel of 12 PRISM-compatible antibodies (Supplementary Table 2 and Fig. 2a). Based on staining intensity, signal specificity, and co-localizations between target markers, we were able to use 3+ specific markers to define a cell's identity (Supplementary Table 2 and Fig. 2a). For each selected antibody, we further compared the staining signal between IF/ICC and PRISM before and after PRISM conjugation (Fig. 2b and Supplementary Fig. 4) to ensure each was still highly selective for its target, and that the fluorescence pattern was consistent and unchanged between IF/ICC control and PRISM signal (Supplementary Fig. 7). We used conventional phalloidin488 and DAPI nuclear stain to align the imaged areas between staining and washing steps.

Newly generated DNA imaging strands were supplemented with published LNA imaging strands[16] for an initial round of 10 DNA-PRISM pairs (Supplementary Table 7 and Supplementary Figs. 4 and 5). Fluorescence signals were suitable for both manual and automated quantification of all ten PRISM markers and two control IF/ICC markers (Supplementary Fig. 6), as they each produced signal that was able to be characterized via our automated CellProfiler pipeline. We compared DNA-PRISM with traditional IF using cross-correlation analysis to map specific PRISM signals to subsequent IF signals in the same culture to validate all 12 markers in human iPSC-derived cortical neurons (Supplementary Fig. 7) and in rat hippocampal neurons validated the tested markers (Supplementary Fig. 8) for appropriate target recognition. The signal to noise ratio was calculated by comparing the average ($N = 6$ samples) signal from the PRISM stain to the average ($N = 6$ samples) of the background signal in the image. A full DNA-PRISM marker panel in rat hippocampal neurons was performed, showing that DNA-PRISM can be used to characterize neural cultures from different organisms, while maintaining specificity of marker staining patterns (Supplementary Fig. 9), including synaptic markers (Supplementary Fig. 11).

Validation and optimization included matching staining patterns in primary rat hippocampal neurons and human stem cell-derived (BJ-SiPs) cortical neurons. We compared control IF/ICC and PRISM by Pearson Correlation analysis with values ranging between 0.65 (synaptic markers) and 0.89 (cytoskeletal markers). Signal-to-noise ratios were at least 25:1 based on pixel intensity [A.U.], and all validated markers minimal off-target binding. We saw some non-specific signal between the validated oligos and

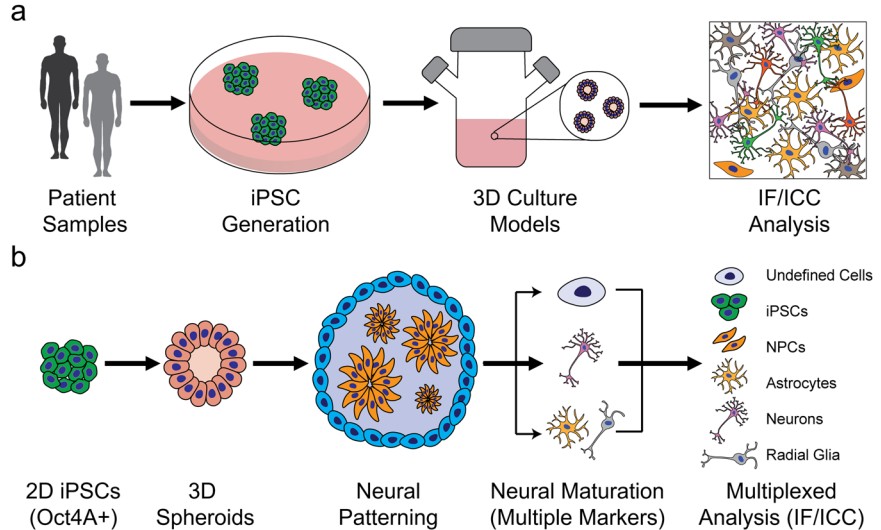

**Fig. 1 Generating stem cell derived disease models and complex neuronal cultures based on patient iPSCs requires careful and extensive validation. a** Process flow to generate disease-specific stem cell lines from patients to model in vitro complex neurodevelopmental and neurodegenerative diseases using 3D spheroid cultures of cortical and motor neurons. **b** Stem cell-derived 3D spheroid neural cultures were dissociated and used to generate high-content data for culture and disease modeling. Cells are subsequently analyzed to generate specific culture breakdowns using PRISM for in-depth characterization of neural cultures via multiplexed staining.

any present cell DNA or RNA transcripts, which is a challenge when imaging nuclear transcription factors, but we were able to further reduce any residual non-specific binding to native DNA and/or RNA with salmon sperm DNA blocking and RNAse treatment.

**PRISM enables high content imaging of over ten neural markers in 2D and 3D iPSC-derived CN and MN cultures.** iPSC-derived CNs and MNs were differentiated in 3D suspension cultures and then dissociated, plated in standard 2D culture wells, maintained for 14 days, and then fixed and stained for analysis. First, BJ-SiPs iPSCs were differentiated into CNs (Supplementary Table 4) and characterized via multiple markers by traditional IF/ICC to confirm their cortical nature (Supplementary Fig. 1). Representative images from the dissociated CN culture illustrate multiple cell subtypes (Fig. 3a). We further characterized in-depth two time points during differentiation of cortical cultures in relation to cell identity (Fig. 3b), where each cell subtype was compared to the list of characterization markers defined in Fig. 2a. The consistency of differentiation and staining patterns were validated by examining at least six independent wells of the same differentiation per timepoint. The 55 and 85 day timepoints were two different differentiation experiments, but started at the same time, as the cells needed to be fixed for imaging. The images used in the analysis presented here were from the same well, and same area. Raw data for the analysis can be found in Supplementary Data 1 for day 55 cultures and Supplementary Data 2 for day 85 cultures. At day 55 of cortical differentiation, there were 84/200 cells that fulfilled our analysis criteria (Fig. 2a), where we were able to positively identify 30% of the analyzed cells as neurons. Synaptic marker staining characterized 100% of identified neurons as excitatory in this earlier culture (VGLUT1+). Immature/inactive astrocytes (CD44+/Vimentin+) made up 18% of the culture and 7% of the cells expressed markers for mature/activated astrocytes (CD44+/GFAP+). We were also able to positively identify both neural progenitor cells (2%, Pax6+) and a large percentage of radial glial cells (36%, Pax6+/Vimentin+). Cells that were present but could not be positively identified represented 7% of the total population. We then followed up with a day 85 differentiation timepoint, where we identified 161/200 cells that fulfilled the criteria for analysis as outlined in the "Methods" section. Of these, 58% were positively

identified as neurons (Tuj1+/MAP2+) and 26% as astrocytes (CD44+/Vimentin+/GFAP+). Neurons were successfully further sub-typed based on synaptic marker expression into either excitatory (52%, VGLUT1+) or inhibitory (3%, VGAT+). The remaining 45% of identified neurons were not strongly associated with either type. Astrocytes were further characterized as either immature/inactive (19%, Pax6−/Vimentin+/CD44−) or mature/active (7%, Pax6-/GFAP+/CD44+). We observed no presence of either neural progenitor (Pax6+) or radial glial cells (Pax6+/Vimentin+). We were unable to classify 16% of cells present, based on our characterization criteria.

A parallel PRISM assay of iPSC-derived MN cultures validated the full panel of conjugated antibodies (Fig. 4) to characterize the MN cultures in a similar manner, which highlights the flexibility of the described PRISM system and may allow for future work in diverse neural disease modeling and characterization of developmental pathways. To validate that the generated neuronal cells are indeed motor neurons, we stained separate wells on the same plate with the motor neuron markers Islet 1 and 200 kD neurofilament protein (Supplementary Fig. 2). Subsequently, PRISM antibodies generated highly specific signals, and the images were then overlaid to generate high-content imaging datasets. We show here that DNA-PRISM antibodies can characterize complex stem cell-derived cultures in a multidimensional manner with the added ability to preserve the spatial relationships between the markers used in our characterization pipeline. Further optimization to improve staining quality by testing antibodies, which are generated to produce high specificity and optimized signal-to-noise ratios in the putative MN cultures in particular, is ongoing.

**Automated pipeline to pair PRISM high-content data analysis with high throughput data generation.** Due to the high volume of raw data that is generated in a PRISM imaging assay, we endeavored to develop an automated platform that could manage staining, imaging, and data analysis in the same pipeline. We first identified key points in our manual PRISM assay data acquisition process, namely buffer exchanges, imager strand incubation, and IF/ICC imaging, and automated them using a BRAVO liquid handler and a PerkinElmer Phenix spinning disk confocal microscope. Next, points of adaptation within the physical assay

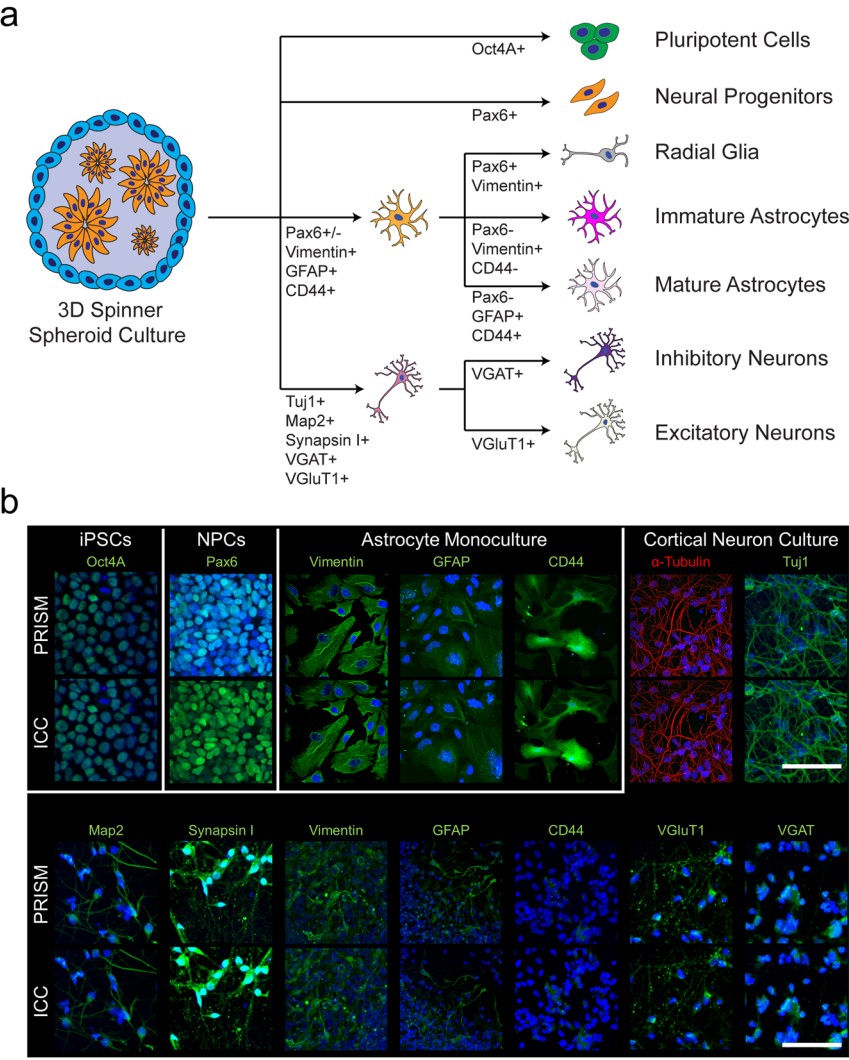

**Fig. 2 Breakdown of cell populations present in iPSC-derived neural cultures and validation staining of marker panel targets used in the in-depth characterization.** A schematic (**a**) of the different cell types in a typical cortical culture that can be tracked via PRISM. The flow chart illustrates a representative readout of the breakdown of cell populations in cortical cultures. **b** ICC versus PRISM images of stem cell derived day 85 cortical neurons to demonstrate staining of the markers used to characterize neural cultures. The Oct4A marker was imaged in the undifferentiated pluripotent state cell culture (iPCS), and the PAX6 marker was imaged in day 14 neural progenitor state cell culture (NPCs) for validation purposes. Scale bars are 100 microns.

steps such as buffer volume, aspiration and dispensing speed, and incubation times were streamlined to improve automation. Slowing buffer aspiration and addition into the wells greatly reduced cell detachment and damage during the twenty consecutive buffer exchanges, which were necessary for a complete PRISM assay. Representative PRISM staining performed with automation (Supplementary Fig. 10a) was comparable to manual PRISM staining and imaging (Fig. 3a). Automation of the imaging pipeline reduced the time to analyze the neural cultures, and since we incorporated automated liquid handling, the overall assay time for the mechanical steps, such as washing, incubation, and oligo exchanges, was drastically reduced compared with the manual assay analog. While the average time for data acquisition, that is, the imaging parameters used by the confocal microscope, was similar between the manual and automated assays, incorporating automated CellProfiler and FIJI analysis pipelines (currently available as Supplemenary Data 3 or at cellprofiler.org) further reduced the analysis time by almost 3-fold, resulting in an overall reduction of assay duration while increasing throughput (Supplementary Fig. 10b).

## Discussion

Recently, high content analysis has been used to characterize multiple neurological cell types and their interactions in vivo and in vitro[30,31], providing insights into cell differentiation and neurological conditions[32–36]. Such work can benefit from a complementary IF method that is not constrained by limitations inherent to traditional antibody-based imaging. In our analysis, we used DNA-conjugated PRISM antibodies to evaluate complex CN and MN cultures derived from stem cell lines in dissociated 2D cultures. In future studies, we envision that PRISM may be used for large scale analysis of neural disease initiation and progression, such as in ALS, focusing on morphology, synaptic make-up, and interactions between neurons and glial cells that are present in these cultures[37].

We used PRISM to perform multidimensional characterization of heterogeneous cell types that have direct pharmacological and disease modeling applications. DNA/DNA or DNA/LNA pair strands with a number of fluorophore modifications are readily available from commercial vendors, making such antibody panels relatively straightforward to generate for a wide range of research

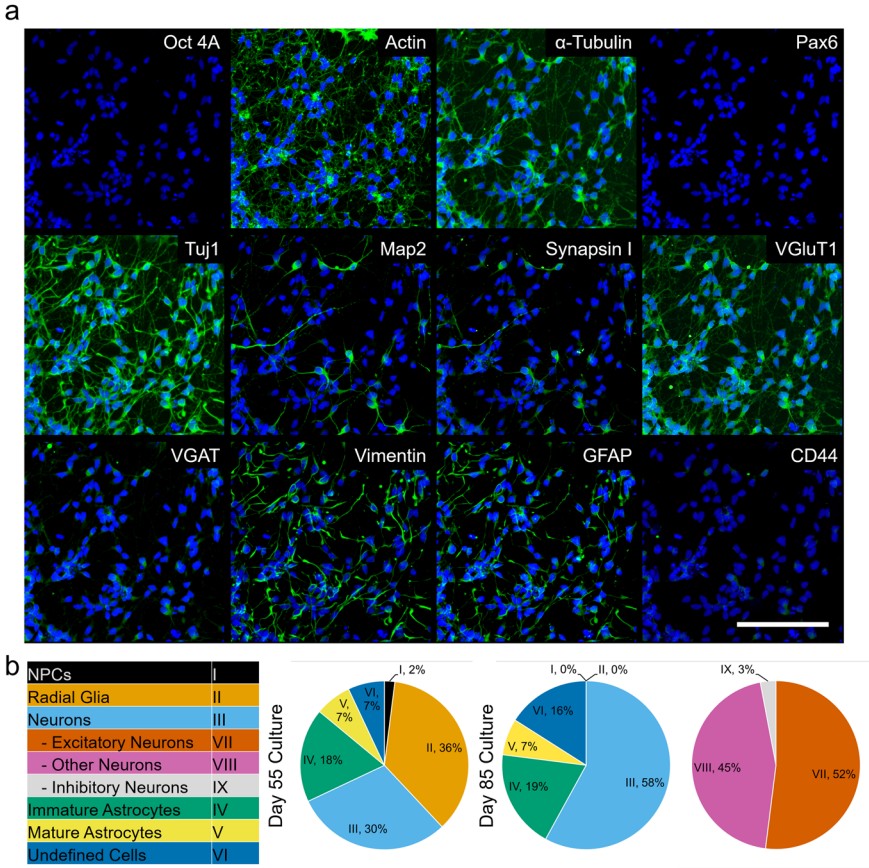

**Fig. 3 High-content analysis of dissociated cultures from hiPSC-derived cortical neurons. a** Representative images from the same area of a 85 days old BJ-SiPs-derived cortical neuron culture, stained with the optimized 12-marker PRISM antibody panel, 14 days post-dissociation into 2D cultures. **b** CellProfiler quantification of cell types at 55 and 85 days in culture respectively. Scale bar is 100 microns.

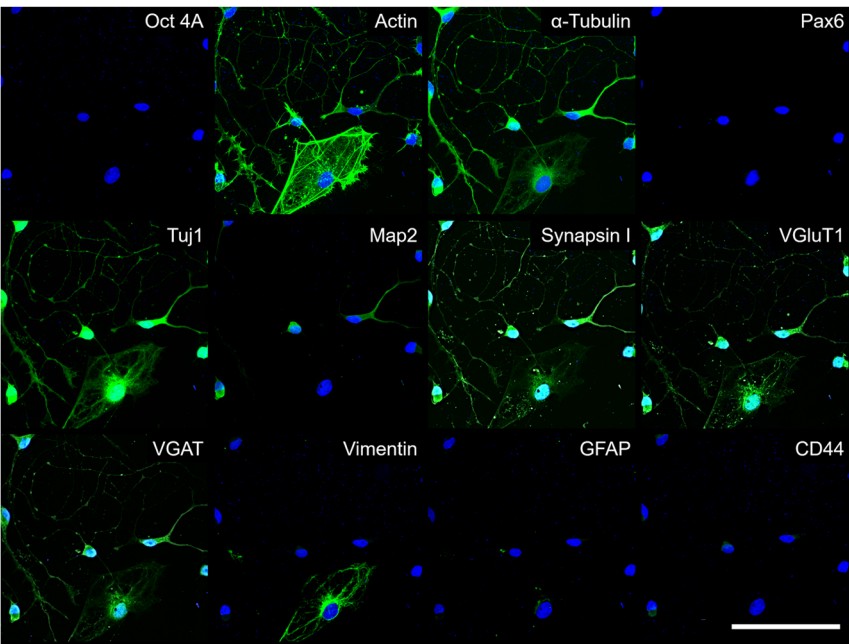

**Fig. 4 High-content IF/ICC analysis of dissociated cultures from hiPSC-derived motor neuron culture.** Representative images from the same area of a 1016A-derived motor neuron culture, stained with the optimized 12-marker PRISM antibody panel after being grown for 21 days as a 3D spheroid, then dissociated into 2D and maintained for 14 days. Scale bar is 100 microns.

applications. Wang et al.[38] reported that DNA-only oligos can have high non-specific binding to native transcripts. LNA provides somewhat higher signal to noise ratio compared to DNA, and it less likely to participate in non-specific binding to native DNA or RNA sequences in the target cell. DNA on the other hand is easier, faster, and cheaper to generate fluorophore conjugated oligos with. Combined with our improved blocking and washing steps, we see similar results between LNA and DNA for most, but not all (e.g., some synaptic and nuclear transcription factors) targets.

To mitigate this issue, we implemented a salmon sperm DNA blocking step, as well as RNase pre-treatment of the fixed cultures when LNA strands were used (see Supplementary Table 3 for detailed protocol). We further chose DNA oligo pairs between 11–12 nucleotides with GC content between 30–40%, which we found to be optimal for reducing background fluorescence while still allowing complete washout or exchange of imaging strands post-acquisition. High melting temperatures of DNA/LNA oligos may push PRISM into a regime where samples need to be heated, even under low salt concentration, in order to release imaging strands. High melting temperature also can cause fluorescence artifacts such as residual signal and non-specific binding, and increased cell detachment due to the need for heating and cooling of the sample, thereby complicating assay automation[26,39,40]. While fluorescence from the PRISM staining was considerably lower than the corresponding signal from conventional secondary antibodies (Supplementary Fig. 4c), it was still well above background levels. This limitation might be overcome with further development of the technique to increase the number of fluorophores on the imager strands, or by using longer exposure times.

The 12 markers that we used in our assay allowed in depth analysis suggesting that disease-specific morphological and gene expression differences could be elucidated. While misclassification of cells is certainly possible, we have endeavored to minimize this by using multiple markers per cell type and stringent exclusion criteria, where a cell must be positive for all of the designed markers in order to be included in the final groups. Combined, this allows us to automatically identify and remove potentially mis-classified cells from further contributions to the analysis. These findings open the door to using PRISM antibodies for drug screening and characterizing in vitro disease models. We further expanded our fluorophore detection lines from one[16] to two, halving the time necessary for imaging and reducing the number of wash steps, which decreased cell detachment, thus improving our multidimensional analysis reproducibility. To take full advantage of the multidimensional data set that was generated, we incorporated in-depth analysis capabilities using a custom CellProfiler pipeline and automated liquid handling to perform the stain and wash steps. While the liquid handler and microscope were not connected for this experiment, so the sample transfer was done manually, we are planning to complete a fully automated pipeline in the future. This pipeline can characterize cell subsets within cultures in general, and in the cortical cultures characterize the localization and density of synaptic markers within defined neurites. PRISM would likely be especially well suited to support studies of polygenic psychiatric conditions (Autism Spectrum Disorder and Schizophrenia), neurodevelopmental (ALS), and neurodegenerative (Parkinson's and Alzheimer's) conditions, which are characterized by multiple contributing genes and protein targets[41–47].

## Methods

**Consent for use of human material**. All human material (iPSC lines) was obtained with informed consent and used with the approval of the Harvard University IRB and ESCRO committees. The BJ-SiPs cell line was derived from a healthy male donor using Sendai virus reprogramming. The 1016A stem cell line was derived from a healthy male donor using Sendai virus reprogramming.

**DNA-PRISM antibody marker selection, conjugation, and validation**. Starting from a list of commonly used neural culture-specific antibodies (Supplementary Table 1), we chose a set of twelve markers to build the PRISM antibody panel (Fig. 2). The antibody panel features housekeeping and structural targets (α-Tubulin, Actin), markers for canonical neural culture characterization (Tuj1, Map2, Synapsin I, GFAP, Vimentin, and CD44), as well as cell-specific markers, including those for identifying cortical (VGAT, VGLUT1) and motor neurons (Islet1), glial cells, which are separated into two subtypes, Radial Glia (Pax6, Vimentin, GFAP, and CD44) and astrocytes (Vimentin, GFAP, and CD44), neural progenitor cells (Pax6), and residual pluripotent cells (Oct4A) that might have been retained post-differentiation. Additionally, glia cells could be seen as expressing synaptic markers if they are actively recycling these in synaptically active neural co-cultures. (Supplementary Table 2). For each antibody, validation was first performed using standard IF/ICC prior to oligo conjugation, which was performed as previously described[16]. Signal to noise ratio for the conjugated antibodies was evaluated as the average fluorescence signal of the PRISM stain ($n = 6$ measurements) in question normalized to an average of the background signal in the imaged area ($n = 6$ measurements).

**Confocal imaging of control markers and PRISM panel**. Cultures were fixed in 4% paraformaldehyde for 15 min at room temperature, then washed in phosphate buffered saline (PBS) and stored at 4 °C until they were stained and imaged. Immediately prior to staining, fixed cultures were quenched in 100 mM glycine in ddH$_2$O for 10 min at room temperature. After quenching, the fixed cells were permeabilized in 0.2% Triton X-100 in PBS for 15 min at room temperature, and blocked in 2% bovine serum albumin (BSA) in PBS supplemented with 1 mg/mL salmon sperm DNA (catalog #D7656; Sigma-Aldrich) for 1 h at 4 °C. For marker validation and optimization of staining conditions, cultures were imaged on a Nikon Ti-E spinning disk confocal microscope. High-content imaging with the validated conjugated antibodies panel was then carried out on a PerkinElmer Opera Phenix high-content confocal microscope. The PRISM imaging sequence was performed as previously described[16], with some modifications. We used all primary antibodies in the panel, both in regular IF/ICC and for PRISM assays, at 1:400 dilution and all secondary fluorophore-conjugated antibodies at 1:1000 dilution for IF. Secondary PRISM-conjugated antibodies were used at 1:200 dilution. We used highly cross-adsorbed secondary antibodies for PRISM to minimize any possible signal cross-talk. For validation of marker staining, we used fluorophore-conjugated donkey secondary antibodies, and these same antibodies were also used without primary targets as antibody-only controls. Finally, we used either AF594 or AF647 fluorophores to conjugate to the PRISM imager strand oligos so that we could image two PRISM-stained markers at the same time rather than just one. The full list of antibodies is provided in Supplementary Table 1, with specific staining protocols for DNA-PRISM, including imaging/other buffers, provided in Supplementary Table 3.

**Induced pluripotent stem cells (iPSCs) maintenance**. Human iPSCs (BJ-SiPs and 1016A) were maintained feeder-free on Matrigel hESC-Qualified Matrix (catalog #354277; Corning) coated plates in StemFlex (catalog #A3349401; ThermoFisher) or mTeSR1 (catalog #85857; STEMCELL Technologies) medium to maintain pluripotency and for expansion. Both media were supplemented with pen/strep (1×; catalog #15140122; Gibco). Media were changed every other day for StemFlex and every day for mTeSR1, with cells passaged when they reached >80% confluence.

**Neural progenitor cells (NPCs) maintenance**. Human NPCs were generated using the STEMdiff SMADi neural induction kit (catalog #08581; STEMCELL Technologies) according to the manufacturer's instructions. Briefly, dissociated iPCSs were seeded as a monolayer on Matrigel-coated plates in the provided neural induction medium (NIM) for 14 days, with daily medium changes. NPC expansion, where needed, was performed in Neural Progenitor Medium (catalog #05833; STEMCELL Technologies) from the same kit. On average, NPCs doubled every 3–5 days.

**Rat primary hippocampal neurons maintenance**. Dissociated rat hippocampal neurons were plated on Matrigel-coated plates and maintained in Neurobasal (NB) Medium (catalog # 21103049 ThermoFisher) supplemented with B27 plus insulin (1×; catalog #17504044 Gibco), non-essential amino acids (1× catalog #11140050 Gibco), and 1× pen/strep for 21 days, with half medium changes every 5 days. We complied with all relevant ethical regulations for animal testing and research at the Broad Institute regarding harvesting these cells. Procedures for rat neuronal culture were reviewed and approved for use by the Broad Institutional Animal Care and Use Committee.

**Human iPSC-derived cortical neuron differentiation and maintenance**. Human cortical neurons (CN) were differentiated as spheroids in a 3D spinner flask based

on previously established protocols[48,49] with some modifications. Pluripotent iPSCs were dissociated into single cell suspensions using Accutase (1×; catalog #07920 STEMCELL Technologies) for 10 min at 37 °C. Cells were then adapted to a spinner flask at a concentration of $1 \times 10^6$ cells/mL in mTeSR1 medium supplemented with rho kinase (ROCK) inhibitor, Y-27632 (10 μM; catalog #SCM075 EMD Millipore) for 2 days. Neural progenitor patterning with the small molecules SB431542 (10 μM; catalog #1614 Tocris), LDN193189 (1 μM; catalog #6053 Tocris) and XAV939 (2 μM; catalog #3748 Tocris) was carried out over the next 4 days, with full medium changes every day. Cells were grown in mTeSR1 supplemented with the above factors for the initial 24 h, then the cells were maintained in Knockout Serum Replacement (15% KSR; Thermo Fisher Scientific) medium for the next 3 days, again supplemented with the above factors. Between days 5–11, we gradually transitioned the cells into NIM as follows. On day 5, 75% KSR Pulse medium was mixed with 25% NIM medium supplemented with SB431542 (10 μM) and LDN193189 (1 μM) and cells were incubated for 2 days. On day 7, 50% KSR medium and 50% NIM medium were supplemented with LDN193189 (1 μM) for another 2 days. On day 9, cells were kept in 25% KSR mixed with 75% NIM medium with LDN193189 (1 μM) for another 2 days. On day 11, the 3D cultures were fully transitioned to NIM medium for 9 days with full medium changes every 3 days. The NIM medium at this point was not supplemented with any additional small molecules. On day 20, spheroids were transitioned to NB medium, supplemented with B27 Supplement (1×; catalog #A3582801 ThermoFisher), N2 Supplement (1×; catalog #17502048 ThermoFisher), brain-derived neurotrophic factor (BDNF; 10 ng/mL; catalog #248-BD-010 Tocris) and glial cell-derived neurotrophic factor (GDNF; 10 ng/mL; catalog #212-GD-010 Tocris), and maintained with full medium changes every 4 days until day 40. At this point, mitotically active cortical progenitors could be cryo-stored or maintained as spheroids for a further 14 days or more (up to 100 days) to generate mature CN for functional analysis, with 50% medium changes every 3 days. For staining and analysis, spheroids were dissociated in 0.25% Trypsin-EDTA (catalog #15575020 Thermofisher) to single cells. The cell suspension was plated into 96-well plates that were previously coated with poly-D-lysine (25 μg/mL) and poly-L-ornithine (25 μg/mL) overnight at 37 °C, and then further coated with Laminin (10 μg/mL; catalog #11243217001 Sigma-Aldrich) for 3 h at 37 °C. These 2D cultures were then matured for an additional 14 days prior to DNA-PRISM marker analysis, with 50% medium changes every 3 days. See Supplementary Table 4 for detailed protocol steps and Supplementary Fig. 1 for representative images of cortical cultures.

**Human iPSC-derived motor neuron differentiation and maintenance**. Motor neurons (MN) were generated using a modified version of previously established protocols[50,51]. iPSC colonies were dissociated into single cells using Accutase. Cells were then seeded into ultralow attachment dishes in mTESR1 medium supplemented with ROCK inhibitor, Y-27632 (10 μM; catalog #1254 Tocris) and basic fibroblast growth factor (FGF-2; 20 ng/mL; catalog #233-FB-010 Tocris) for the first 24 h to allow for embryoid body (EB) formation. The following day, ROCK inhibitor was removed, and fresh mTESR1 was added to the cultures. Forty-eight hours after EB aggregation, cells were switched to MN Differentiation Medium: Advanced DMEM/F-12 (catalog #12634010 ThermoFisher) and NB Medium (50:50 v/v), 1× N-2 supplement, 1× B27 + Insulin, 1× GlutaMAX (catalog #A1286001 ThermoFisher), 1× pen/strep, and 0.1 mM 2-mercaptoethanol (catalog #31350010 ThermoFisher). To specify neural patterning, dual SMAD inhibition was used with small molecules SB431542 (10 μM) and LDN193189 (100 nM) from day 0 to day 6 of differentiation. From day 0 to day 4 the glycogen synthase kinase 3 inhibitor, CHIR99021 (3 μM; catalog #4423 Tocris) was added to increase the population of Olig2 positive MN progenitors. Beginning on day 2, MN specification was induced with 1 μM All-trans Retinoic Acid (at-RA; catalog #0695 Tocris) and 1 μM Smoothened Agonist (SAG; catalog #4366 Tocris) until day 16. The γ-secretase inhibitor, (2S)-N-[(3,5-Difluorophenyl)acetyl]-L-alanyl-2-phenyl]glycine 1,1-dimethylethyl ester (DAPT, catalog #2634 Tocris) was used at 10 μM in conjunction with neurotrophic factors BDNF and GDNF (10 ng/mL each) starting on day 8 until day 16 of differentiation, with 50% medium changes every 2 days. Spheroids were then dissociated with a solution containing 0.25% Trypsin-EDTA and DNAse (25 μg/mL; catalog #18047019 ThermoFisher), and plated onto poly-L-ornithine (25 μg/mL; catalog #P2533-10MG Sigma-Aldrich), Fibronectin (10 μg/mL; catalog #11051407001 Sigma-Aldrich), and Laminin (10 μg/mL)-coated plates in medium supplemented with BDNF and GDNF (10 ng/mL each). See Supplementary Table 5 for detailed protocol steps and refer to Supplementary Fig. 2 for differentiation protocol outline and representative images of MN cultures.

**Human iPSC-derived astrocyte differentiation and maintenance**. Human NPCs were expanded as progenitors and then seeded at ~20% confluence onto Matrigel-coated tissue culture plates. Commercial astrocyte medium (catalog #1801 Sciencell) was used to differentiate NPCs as previously described[52,53]. Briefly, starting with ~40% confluent NPC cultures, astrocyte medium was changed every 3–4 days and cells were passaged at 1:10 ratio when they reached ~80–90% confluence for the first 28 days. Immature astrocytes generated by this method express multiple canonical glial markers and were further matured via small molecules or exposure to FBS/Matrigel into GFAP+ cells[54]. See Supplementary Fig. 3 and Supplementary Table 6 for detailed protocol and representative images.

**Design of imaging strands for PRISM antibody conjugation**. We designed new DNA imaging strands and used previously published LNA sequences to generate stable PRISM pairs by varying length and GC content of the DNA oligos (Supplementary Table 7). This was done in order improve signal specificity and minimize strand decoupling due to low melting temperature, ensuring that salt concentraton variation was the dominant means to cycle fluorescent oligos.

**Automated pipeline for antibody staining, imaging, and characterization of cortical cultures**. To validate PRISM antibodies for automation suitability, we performed IF imaging on two high content plate confocal microscopes, the PerkinElmer Opera Phenix and the Molecular Devices ImageXpress. Once antibody signal was confirmed, IF staining of cortical cultures (Supplementary Fig. 10) was adapted to 96-well plates and was partially automated using a BRAVO liquid handler (Agilent); dispensing 100 μL of either buffer, antibody, or PRISM reagent per well (See Supplementary Table 3 for details on buffer compositions and dilutions). The system was programmed to perform an initial rinse with Wash Buffer, then each well was aspirated and 100 μL of fresh Imaging Buffer with PRISM imaging strands was added and incubated at room temperature for 10 min. Plates were then rinsed three times with Imaging Buffer and imaged on a PerkinElmer Opera Phenix at 20×, with the 405 nm laser for Hoechst nuclear stain (IF), 488 nm laser for Actin (IF), and 590 nm and 647 nm lasers for PRISM antibodies. Exposures were either 300 ms (PRISM antibodies) or 150 ms (ICC controls). Post-imaging, each well was washed three times with Wash Buffer and incubated for 5 min between washes to eliminate residual PRISM imaging strand signal. Control imaging was performed after the third rinse to confirm removal of imaging strands prior to subsequent imaging strand addition and imaging rounds. These steps were repeated until all PRISM antibodies in the panel were imaged. Further automation was achieved by incorporating robotics to move the plates for confocal imaging to the PerkinElmer Opera Phenix high content platform at 60x magnification, where each well was imaged with numerical aperture (NA) of 1.15 in a 4 × 4 tile pattern, covering roughly 10% of the well area, or 0.03 cm². Based on cell type-specific markers that have been validated in the field (Supplementary Table 2), we then built up a query algorithm to identify each cell subtype we were interested in within the differentiated cultures. Only cells that were positive for all the required markers were included in the analysis groups within each sub-type. A detailed breakdown of cellular characterization is shown in Fig. 2a.

Data were analyzed using custom CellProfiler (cellprofiler.org) and FIJI (https://fiji.sc/) pipelines for post-processing and image analysis of high-content PRISM data (reference code is available as Supplementary File). Each image from the PRISM assay was aligned in sequential imaging rounds using the nuclear (Hoechst 33342; 405 nm) and actin (Phalloidin; 488 nm) channels. The aligned signal from each marker was then overlaid in a composite image that was then analyzed for protein co-localization within the cells, expression patterns of specific markers in the culture and levels of protein expression. Only cells that were present in all image sequences of an experiment, based on the overlays, were fed to the CellProfiler analysis pipeline to characterize the neural cultures. Additionally, we also took into account structural parameters (e.g., filament for Tij1 or puncta for Synapsin) in addition to fluorescence intensity to improve classification by removing possible artifacts. Sampling data from 12 markers, i.e., five probe addition/imaging sequences, allowed us to generate multi-stain identity of cellular populations that were designated as neurons (cortical or motor), astrocytes, neural stem cells, etc., building cellular composition profiles of the tested iPSC-derived cortical and motor neuron cultures.

**Statistics and reproducibility**. We used at least three independent stem cell-based neural differentiations (three for cortical and three for motor neurons) for our DNA-PRISM imaging. We performed staining on each culture independently and with freshly conjugated antibodies to ensure reproducibility. For the CellProfiler analysis we used at least 100 cells per condition. Statistical analyses performed here included standard error of the mean and standard deviation.

**Reporting summary**. Further information on research design is available in the Nature Research Reporting Summary linked to this article.

## Data availability
The data generated for this manuscript and/or used to support the findings of this study are presented in this manuscript and its supplementary files.

## Code availability
CellProfiler code can be accessed at cellprofiler.org, or by directly contacting the corresponding author and it is under no restrictions to access it.

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

## Acknowledgements

Funding from the NIH BRAIN Initiative Award 1U01MH106011 to M.B., NIH R01-MH112694 to M.B. and M.L.T., and the NSF Physics of Living Systems 1707999 to M.B. and M.L.T. is gratefully acknowledged. Additional funding to M.B., A.E.C., and M.L.T. came from Skoltech 1911/R. M.B., A.E.C., M.L.T., B.C., L.L.R., and H.S.A acknowledge funding from the Stanley Center for Psychiatric Research. L.L.R. also received funding from the Harvard Stem Cell Institute. We also acknowledge the Whitehead Institute W.M. Keck Microscopy Facility, for technical support. Support for this research was provided by a core center grant (P30-ES002109) from the National Institute of Environmental Health Sciences, National Institutes of Health.

## Author contributions

M.L.T., L.L.R., and M.B. conceived of and designed the DNA-PRISM study. M.L.T. implemented the antibody conjugation protocols for DNA-PRISM and designed and implemented the DNA-PRISM experiments. M.L.T., H.S.A., B.A.C., and A.E.C. analyzed the DNA-PRISM data. H.S.A, B.A.C., and A.E.C. designed the CellProfiler analysis pipelines. A.O'N. prepared stem cell differentiated cortical and motor neuronal cultures

and helped M.L.T. with culture fixation and preparation. M.L.T. implemented the DNA-PRISM fixation, blocking, staining, probe exchange, and imaging protocols. M.L.T. adapted the DNA-PRISM protocol for dual fluorophore imaging. M.B. designed and supervised the overall project including DNA-PRISM. L.L.R. supervised the stem cell cortical and motor neuron portion of the project. M.L.T., L.L.R., A.E.C., and M.B. interpreted the DNA-PRISM results and CellProfiler pipelines. M.L.T., A.O'N., L.L.R., and M.B. wrote the manuscript. All authors commented on the manuscript.

## Competing interests

The authors declare no competing interests.
