## [Transparent Peer Review File · Communications Biology]

Reviewers' comments:

Reviewer #1 (Remarks to the Author):

In this paper Tomov et al. developed a PProbe-based Imaging for Sequential Multiplexing (PRISM) to allow multiplexed imaging using oligonucleotide probes to analyze iPSC-derived cortical and motor neuron cultures development by analyzing in the same sample up to ten protein targets. The data presented showed the possibility to analyze the expression of several proteins using multiple fluorescent staining with DNA conjugated antibodies. The majority of the experiments aimed to set up and optimize the use of commercial antibodies with the DNA fluorescent probe.

The data presented are interesting and useful for the laboratories that desire to develop these imaging techniques.

Minor point:

- To better visualize the staining of synaptic proteins the authors should provide some images of neurons stained with synaptic antibodies at higher magnification.

Reviewer #2 (Remarks to the Author):

This manuscript describes efforts to validate the use of PRISM in human iPSCs and iPSC-derived cultures of cortical and motor neurons. PRISM is a recently developed strategy to use oligonucleotide sequences complementary to LNA or DNA to generate a fluorescent readout that avoids the technical challenges of repetitive stripping, staining and imaging with traditional IF/ICC for multiplexing of numerous markers of cell identity in a single sample. Overall, this is a nicely illustrated manuscript that shows cell-type specificity using PRISM in human iPSC-based cultures. The comparison of IF/ICC and before and after PRISM conjugation allows the reader to see the specificity and intensity of the markers on the same sections, which provides important data when evaluating the utility of this approach. Likewise, the table listing the successful, not successful and ambiguous results to convert antibodies into PRISM antibodies is very helpful and the information will be of value to the community and the inclusion of detailed information about the negative results is appreciated. Overall, although this study is an incremental advance over the previous PRISM studies, it does show the efficacy in human iPSC-derived cultures and would be of interest to the field. Although the number and generally high quality of images was appreciated, there were several omissions of details and explanations in the text that made the manuscript more difficult to parse than necessary. A few specific comments and suggestions are outlined below:

Discussion of the LNA vs DNA-only approach and potential background from DNA-only oligos was warranted, but the images shown in SI Figure 6 would still suggest greater specificity, if not intensity, of the LNA imaging strands. And this topic comprises the bulk of the discussion but is not mentioned previously in the text. The rationale for this comparison should have been made earlier.

Some of the details of the cell type specification are missing. In the 4th paragraph of results (lines 108+), it is mentioned that 84 cells meet "analysis criteria", but out of approximately how many cells/cultures/repetitions and what criteria exactly? Expressing one or more markers of a particular cell type? How are transitional neurons defined?

It's not clear why Figure 4 is a main figure, if the motor neuron PRISM antibodies are included in the 12 antibody panel, although they were successfully converted according to SI Table 1.

For the automated image analysis, an explicit comparison should be made to manual imaging. Also, the text states that the time for data acquisition was the same for manual and automated, but that doesn't appear to match the quantification in Figure SI 10b.

It is a little confusing at several points whether the authors are referring to 3D organoid/spheroid cultures or 2D cultures. For example in Figure 1, "3D neural cultures" is used in the graphical labels in 1A, but in the legend for 1B and vice versa for "3D spheroids". A clearer description and consistent labeling of the 2D vs 3D culturing throughout the figures and manuscript would help readers. And if the data from 3D organoid sections were collected, it should have been shown in one of the figures given the emphasis throughout on the utility of this approach for both types of cultures.

The different human iPSC lines that are listed in the methods are sometimes not specified throughout the text and figures and it's unclear if the ALS patient line was used at all and if there were any differences observed.

There were a few grammatical errors/typos throughout (e.g. lines 86- 88, 551)

Reviewer #3 (Remarks to the Author):

The manuscript by Tomov et al. presents the application of highly multiplexed fluorescence microscopy to classify the cell type in fixed, cultured cells. This is used to observe the differentiation of induced pluripotent stem cells (iPSC) to cortical/motor neurons and associated cell types in 2D and 3D model cultures.

While the general approach of PRISM is similar to exchange DNA-PAINT, the method is modified to allow for a diffraction limited imaging with low fluorescent background and therefore high signal-to-noise ratio.

Compared to the previous publication of PRISM by the Bathe Lab (<https://www.nature.com/articles/s41467-019-12372-6>), this manuscript focuses on the determination of cell types and automation of measurement and data analysis which will be relevant for applications in the context of disease research and drug screening.

The presented data support the validity of the approach and are in general convincing. However, the presentation of the results require some clarifications. It is great that the details of the antibody screening, including the non-working ones, are provided and it would be desirable to see the same level of detail on the implementation of the automation and data analysis, since these are key advances of the presented manuscript.

Detailed comments:

1: The introduction should provide more references to alternatives for highly multiplexed imaging: The work by Klevanski et al. even achieved super-resolution with 15 different targets with a more conventional, but fully automated, bleaching/stripping and relabelling approach. <https://www.nature.com/articles/s41467-020-15362-1>

2: Line 38: "Fluorescence-based antibody staining of target markers [...] is limited by the conventional spectral limit of fluorophores to imaging four targets simultaneously. One technique to overcome this limitation ...". There are several techniques capable of overcoming this limitation, e.g. by including lifetime information (<https://www.nature.com/articles/nmeth.3740>) or hyper-spectral information (<https://doi.org/10.1364/OPTICA.389982>), but PRISM is not able to image more than four targets simultaneously as it is a sequential technique.

3: The first part of the results section, especially the second and third paragraphs (line 82-100), is hard to read and feels in part more like a list of supplemental material. It would help if the results were described/interpreted. Maybe the following questions/comments help to achieve that:

What is the difference between the DNA- and LNA-PRISM images?

What is the result of the quantification (line 84)?

After mentioning "that the fluorescence pattern was consistent and unchanged" (line 80) it seems redundant to confirm that the PRISM signal is significantly above background (line 86).

For me, it also makes more sense to skip the cross-correlation analyses here (line 87) and refer to the underlying dataset when giving the result in line 95.

The whole paragraph starting in line 93 is one sentence and in the end the references are not clear any more:

"... minimal off-target binding and non-specific signal between the validated oligos and any present cell DNA or RNA transcripts, which is necessary [...] to reduce non-specific binding to native DNA and/or RNA ...". Minimal non-specific signal is necessary to reduce non-specific binding?

4: SI Tab 2: It is not apparent what +/- is exactly encoding. Only found in healthy/diseases cells? This should be clarified in the caption.

5: SI Fig 6 (Line 535): "Signal strength comparisons between LNA-PRISM and DNA-PRISM imaging strands was done on three different confocal microscopes ..." It should be mentioned whether the results were consistent across the three microscopes.

Are the error bars in the bottom panel the standard deviation of the mean? Please specify.

6: SI Fig 7: The actin is always stained with phalloidin, so PRISM and ICC are here the same? The caption defines ICC Stain as secondary antibody staining.

Pax6 is only visible in the ICC Stain. Is that a sensitivity problem?

7: Applies to all microscopy images: What is the difference between green and red, e.g. alpha-tubulin is sometimes green, sometimes red? Is this supposed to convey any information about the fluorophore used?

How are the intensities scaled? Is it the same scaling across all images within one panel?

How is the scaling? Please provide a scale bar or state the size of the field of view.

8: Fig 2a: From the schematic it is not clear what the difference between translational neurons and inhibitory/excitatory neurons is. Also, what kind of condition is Pax6+/-? Is there an implicit minus on all non-listed markers?

Fig 2b: What culture was used for the lower part of the panel?

9: Fig 3: Is there a reason the images from day 71 but the analysis from days 55 and 85 are shown? Maybe it can be made clear (graphically) that type III is subdivided into types VII, VIII and IX.

It is not a must, but an image of the cells with an overlay indicating the classification would be a nice addition.

9: SI Tab 7: The direction of the sequences is not stated. For the LNA the sequences of the docker and imager strand seem to have the same direction (5'->3'?) and for the DNA opposite directions. Also, at which end was the fluorophore/antibody attached?

10: Several methodical details, required for replicating the experiments and analysis are missing:

Details of the imaging: The dwell time is stated, however the pixel-size and scan area are missing.

What are the details of the automated measurement procedure: Are the liquid handler and microscope connected or is it a manual sample transfer? What causes the long dead time (47%)?

Details of classification: Only based on the integrated intensity or is the structure considered, e.g. to exclude artifacts? How are the intensity thresholds set to decide if a marker is present.

How is the signal-to-noise ratio (line 96) defined?

Some of these points might also become clear, if the authors would include their CellProfiler pipeline.

11: Similar to the results section, it is not clear from the discussion (line 163-177) what the

differences/advantages of LNA or DNA are.

12: In line 185, it becomes apparent for the first time that two (PRISM) colours are used in parallel. This should also be mentioned in the results/methods section.

13: Potential miss-classifications of cells are not discussed. Please comment on the accuracy of the classification.

14: The abstract suggests that the presented technique is applied to 3D spheroids and organoid sections and the discussion confirms that "3D organoid sections (data not shown)" (line 159) have been analysed. However only data from (dissociated) 2D cultures are presented. The claim that 3D samples can be analysed should be either supported by data or clearly marked as future perspective.

15: line 297: "assuming that's what you used". Please check.

Referee expertise:

Referee #1: neuronal differentiation of neural progenitors from hiPSCs

Referee #2: iPSCs, neurogenesis, organoids

Referee #3: multiplexed imaging, DNA PAINT

Reviewers' comments:

Reviewer #1 (Remarks to the Author):

In this paper Tomov et al. developed a PProbe-based Imaging for Sequential Multiplexing (PRISM) to allow multiplexed imaging using oligonucleotide probes to analyze iPSC-derived cortical and motor neuron cultures development by analyzing in the same sample up to ten protein targets. The data presented showed the possibility to analyze the expression of several proteins using multiple fluorescent staining with DNA conjugated antibodies. The majority of the experiments aimed to set up and optimize the use of commercial antibodies with the DNA fluorescent probe.

The data presented are interesting and useful for the laboratories that desire to develop these imaging techniques.

Minor point:

- To better visualize the staining of synaptic proteins the authors should provide some images of neurons stained with synaptic antibodies at higher magnification.

A: We thank the reviewer for their comments and helpful suggestion, we have added an additional SI Figure 11, where we provide zoom-ins of the synaptic protein staining in both hiPSC-derived and rat neurons.

Reviewer #2 (Remarks to the Author):

This manuscript describes efforts to validate the use of PRISM in human iPSCs and iPSC-derived cultures of cortical and motor neurons. PRISM is a recently developed strategy to use oligonucleotide sequences complementary to LNA or DNA to generate a fluorescent readout that avoids the technical challenges of repetitive stripping, staining and imaging with traditional IF/ICC for multiplexing of numerous markers of cell identity in a single sample. Overall, this is a nicely illustrated manuscript that shows cell-type specificity using PRISM in human iPSC-based cultures. The comparison of IF/ICC and before and after PRISM conjugation allows the reader to see the specificity and intensity of the markers on the same sections, which provides important data when evaluating the utility of this approach. Likewise, the table listing the successful, not successful and ambiguous results to convert antibodies into PRISM antibodies is very helpful and the information will be of value to the community and the inclusion of detailed information about the negative results is appreciated. Overall, although this study is an incremental advance over the previous PRISM studies, it does show the efficacy in human iPSC-derived cultures and would be of interest to the field. Although the number and generally high quality of images was appreciated, there were several omissions of details and explanations in the text that made the manuscript more difficult to parse than necessary. A few specific comments and suggestions are outlined below:

We appreciate the overall positive assessment of this work by the reviewer, their careful reading of the manuscript, and their helpful comments and suggestions below.

Discussion of the LNA vs DNA-only approach and potential background from DNA-only oligos was warranted, but the images shown in SI Figure 6 would still suggest greater specificity, if not intensity, of the LNA imaging strands. And this topic comprises the bulk of the discussion but is not mentioned previously in the text. The rationale for this comparison should have been made earlier.

A: We have expanded the rationale for using DNA and how these DNA oligos compare to LNA oligos in the Introduction (lines 53-58) and in the Methods (lines 364-366). Briefly, we were able to reduce the most significant issues with pure DNA oligos, such as non-specific binding to complementary strands and non-specific transcript binding in this paper, which makes PRISM technology more straightforward to adopt and use.

Some of the details of the cell type specification are missing. In the 4th paragraph of results (lines 108+), it is mentioned that 84 cells meet “analysis criteria”, but out of approximately how many cells/cultures/repetitions and what criteria exactly? Expressing one or more markers of a particular cell type? How are transitional neurons defined?

A: We thank the reviewer for this comment and have clarified in the text (lines 132-148). Specifically, we have added that the total number of cells that were fed into the characterization pipeline was 200 per analyzed differentiation timepoint. We have also clarified in the same section that cell characterization was based on expression of all markers for the specific cell type as listed in Figure 2a. By exclusion then, transitional neurons are defined as cells that do not fit into the defined cell types.

It's not clear why Figure 4 is a main figure, if the motor neuron PRISM antibodies are included in the 12 antibody panel, although they were successfully converted according to SI Table 1.

A: We have clarified in the text (lines 149-152) that the motor neuron culture is included in the main manuscript to demonstrate that the generated PRISM antibody panel is not limited to only cortical neurons, but can also be used to examine in detail other neural cultures. Because the study of neurological and neurodegenerative conditions is increasingly relying on *in vitro* models that are limited by the ability to thoroughly characterize them via fluorescence, we felt that highlighting this was an important point to stress in the main body of the manuscript.

For the automated image analysis, an explicit comparison should be made to manual imaging. Also, the text states that the time for data acquisition was the same for manual and automated, but that doesn't appear to match the quantification in Figure SI 10b.

A: We have clarified the section regarding image analysis and have expanded it with an explicit comparison between automated and manual imaging. We have also clarified that "data acquisition" only refers to the step where images are acquired by the confocal system (lines 172-181).

It is a little confusing at several points whether the authors are referring to 3D organoid/spheroid cultures or 2D cultures. For example in Figure 1, "3D neural cultures" is used in the graphical labels in 1A, but in the legend for 1B and vice versa for "3D spheroids". A clearer description and consistent labeling of the 2D vs 3D culturing throughout the figures and manuscript would help readers. And if the data from 3D organoid sections were collected, it should have been shown in one of the figures given the emphasis throughout on the utility of this approach for both types of cultures.

A: We have clarified throughout the manuscript that the differentiation of stem cell-derived neurons was done in 3D spheroids, which were then dissociated into a 2D culture and allowed to reform neural connections prior to fixation and PRISM staining.

The different human iPSC lines that are listed in the methods are sometimes not specified throughout the text and figures and it's unclear if the ALS patient line was used at all and if there were any differences observed.

A: We have removed mentions of the ALS patient line from the manuscript.

There were a few grammatical errors/typos throughout (e.g. lines 86- 88, 551)

A: We thank the reviewer for their thorough read and have done our best to correct the errors and typos throughout the manuscript.

Reviewer #3 (Remarks to the Author):

The manuscript by Tomov et al. presents the application of highly multiplexed fluorescence microscopy to classify the cell type in fixed, cultured cells. This is used to observe the differentiation of induced pluripotent stem cells (iPSC) to cortical/motor neurons and associated cell types in 2D and 3D model cultures.

While the general approach of PRISM is similar to exchange DNA-PAINT, the method is modified to allow for a diffraction limited imaging with low fluorescent background and therefore high signal-to-noise ratio.

Compared to the previous publication of PRISM by the Bathe Lab (<https://www.nature.com/articles/s41467-019-12372-6>), this manuscript focuses on the determination of cell types and automation of measurement and data analysis which will be relevant for applications in the context of disease research and drug screening.

The presented data support the validity of the approach and are in general convincing. However, the presentation of the results require some clarifications. It is great that the details of the antibody screening, including the non-working ones, are provided and it would be desirable to see the same level of detail on the implementation of the automation and data analysis, since these are key advances of the presented manuscript.

We appreciate the positive assessment of this work by the reviewer.

Detailed comments:

1: The introduction should provide more references to alternatives for highly multiplexed imaging: The work by Klevanski et al. even achieved super-resolution with 15 different targets with a more conventional, but fully automated, bleaching/stripping and relabelling approach. <https://www.nature.com/articles/s41467-020-15362-1>

A: We have updated the introduction with relevant papers that show alternatives for highly multiplexed imaging.

2: Line 38: "Fluorescence-based antibody staining of target markers [...] is limited by the conventional spectral limit of fluorophores to imaging four targets simultaneously. One technique to overcome this limitation ...". There are several techniques capable of overcoming this limitation, e.g. by including lifetime information (<https://www.nature.com/articles/nmeth.3740>) or hyper-spectral information (<https://doi.org/10.1364/OPTICA.389982>), but PRISM is not able to image more than four targets simultaneously as it is a sequential technique.

A: We have clarified the statement in the manuscript, incorporating the reviewer's suggestion (lines 40-51). Specifically, we clarify that PRISM is still limited to acquiring up to 4 channels at a time, but by cycling the fluorophore-conjugated oligos, more markers can be imaged and characterized.

3: The first part of the results section, especially the second and third paragraphs (line 82-100), is hard to read and feels in part more like a list of supplemental material. It would help if the results were described/interpreted. Maybe the following questions/comments help to achieve that:

A: We highly appreciate the suggestions on how to improve this section of the manuscript and have incorporated most of their suggestions. Detailed responses are below.

What is the difference between the DNA- and LNA-PRISM images?

A: The images generated with DNA-PRISM and LNA-PRISM show that the two oligo types can both generate comparable fluorescent results in our technique; thus they can be used as supplementary to one another, especially when there are markers that might generate off-target artifacts, such as nuclear transcription factors.

What is the result of the quantification (line 84)?

A: We have added a clarification that the quantification result was signal from the PRISM stain that was able to be picked up and characterized in our CellProfiler pipeline (lines 96-97).

After mentioning "that the fluorescence pattern was consistent and unchanged" (line 80) it seems redundant to confirm that the PRISM signal is significantly above background (line 86).

A: We have edited the redundant parts for clarity and brevity (line 96).

For me, it also makes more sense to skip the cross-correlation analyses here (line 87) and refer to the underlying dataset when giving the result in line 95.

A: We have clarified the cross-correlation analysis role as an important validation step in antibody marker selection because there were makers that gave high signal with PRISM, but were not binding to the appropriate target when compared with regular IF (lines 110-118).

The whole paragraph starting in line 93 is one sentence and in the end the references are not clear anymore:

"... minimal off-target binding and non-specific signal between the validated oligos and any present cell DNA or RNA transcripts, which is necessary [...] to reduce non-specific binding to native DNA and/or RNA ...". Minimal non-specific signal is necessary to reduce non-specific binding?

A: We have edited the paragraph in question for clarity (lines 110-118).

4: SI Tab 2: It is not apparent what +/- is exactly encoding. Only found in healthy/diseases cells? This should be clarified in the caption.

A: We have clarified the SI Table 2 in the legend (lines 784-786) and in the text (lines 252-253) that human glia cells could be seen as expressing synaptic markers, due to their role in harvesting and recycling these when co-cultured with functional neurons.

5: SI Fig 6 (Line 535): "Signal strength comparisons between LNA-PRISM and DNA-PRISM imaging strands was done on three different confocal microscopes ..." It should be mentioned whether the results were consistent across the three microscopes. Are the error bars in the bottom panel the standard deviation of the mean? Please specify.

A: We have clarified the figure legend (lines 710-713). Briefly, the results were consistent across the three microscopes and the error bars in the bottom panel represent the standard error of the mean from three experiments performed using the same microscope.

6: SI Fig 7: The actin is always stained with phalloidin, so PRISM and ICC are here the same? The caption defines ICC Stain as secondary antibody staining. Pax6 is only visible in the ICC Stain. Is that a sensitivity problem?

A: This statement has been clarified in the text (lines 97-100) and in the figure legend (lines 728-731). Briefly, the actin stain is used as one of the reference channels that in combination with the nuclear marker allows for precise alignment of the stained area to generate the multi-marker PRISM cell characterization. The lack of Pax6 staining is due to a sensitivity with this specific antibody.

7: Applies to all microscopy images: What is the difference between green and red, e.g. alpha-tubulin is sometimes green, sometimes red? Is this supposed to convey any information about the fluorophore used?

How are the intensities scaled? Is it the same scaling across all images within one panel?

How is the scaling? Please provide a scale bar or state the size of the field of view.

A: There is no significance to the green vs red color for alpha-tubulin. In Figure 2b, the same cells were imaged for both alpha-tubulin and Tuj1, so to distinguish them in the pictures, one was false-colored green and the other was false-colored red. The intensity is the same across all images within each figure. We have also added scale bars to each figure where appropriate.

8: Fig 2a: From the schematic it is not clear what the difference between translational neurons and inhibitory/excitatory neurons is. Also, what kind of condition is Pax6+/-? Is there an implicit minus on all non-listed markers?

Fig 2b: What culture was used for the lower part of the panel?

A: We have rephrased “transitional neurons” into “other neurons” to eliminate confusion. In the text (lines 250-251), we have clarified that cells positive for Vimentin, GFAP, and CD44 can be also either Pax6 negative, characterized as astrocytes, or Pax6 positive, characterized as radial glia. We have also elaborated further in the figure legend for 2b on the origin of the cells (day 85 stem cell derived cortical neurons).

9: Fig 3: Is there a reason the images from day 71 but the analysis from days 55 and 85 are shown? Maybe it can be made clear (graphically) that type III is subdivided into types VII, VIII and IX.

It is not a must, but an image of the cells with an overlay indicating the classification would be a nice addition.

A: The figure legend has been updated to the correct day, which is 85 at time of imaging. The day 71 cultures refer to the time of spheroid dissociation to replace as 2D culture for analysis 14 days later. Because this is confusing, it was simplified in the figure legend (line 660) to just “Day 85”. We have further updated the Neuron (type III) subdivisions in the Figure 3b panel. After careful consideration, we decided that overlaying the classification of the cells over the images would be too busy in the figure and confound the results, rather than clarify them.

9: SI Tab 7: The direction of the sequences is not stated. For the LNA the sequences of the docker and imager strand seem to have the same direction (5'→3') and for the DNA opposite directions. Also, at which end was the fluorophore/antibody attached?

A: We have clarified these issues in the SI Table 7 legend (lines 814-815).

10: Several methodical details, required for replicating the experiments and analysis are missing: Details of the imaging: The dwell time is stated, however the pixel-size and scan area are missing.

A: We have included details on the numerical aperture (NA) and scan area in the Methods section (lines 387-389)

What are the details of the automated measurement procedure: Are the liquid handler and microscope connected or is it a manual sample transfer? What causes the long dead time (47%)?

A: We have expanded our Discussion section to address these comments (lines 229-232). Briefly, the liquid handler and microscope were not connected for this experiment, so the sample transfer was done manually. We are planning to complete a fully automated pipeline in the future. The long dead time is caused by the mechanical transitions of the liquid handler robot, such as micropipette tips pick-up or drop-off, aspiration head movement, buffer aspiration and expulsion, etc., and the microscope, where laser lines need to be swapped to image the 4 fluorophores in the sample sequentially.

Details of classification: Only based on the integrated intensity or is the structure considered, e.g. to exclude artifacts? How are the intensity thresholds set to decide if a marker is present? How is the signal-to-noise ratio (line 96) defined?

A: The classification methodology has been clarified in the methods section (lines 401-405). Briefly, we are also taking into account structural parameters (e.g., filament for Tij1 or puncta for Synapsin) in addition to fluorescence intensity to improve classification by removing possible artifacts. We have further defined the signal-to-noise ratio as the average fluorescence signal of the PRISM stain in question normalized to an average of the background signal in the imaged area (lines 111-114 and lines 253-257).

Some of these points might also become clear if the authors would include their CellProfiler pipeline.

A: The CellProfiler pipeline is currently available upon request from the Carpenter group and will be included on their website (cellprofiler.org) and on the Bathe lab's GitHub (<https://github.com/lcbb>) after this publication.

11: Similar to the results section, it is not clear from the discussion (line 163-177) what the differences/advantages of LNA or DNA are.

A: We have further elaborated on the difference between LNA and DNA in the discussion section (lines 194-200). Briefly, LNA provides somewhat higher signal-to-noise ratio compared to DNA, and it is less likely to participate in non-specific binding to native DNA or RNA sequences in the target cell when sequences are designed appropriately. DNA on the other hand is easier, faster, and cheaper to generate fluorophore conjugated oligos with, as well as to design complementary sequences for given extensive history of DNA primer design. Combined with our improved blocking and washing steps, we see similar results between LNA and DNA for most, but not all (e.g., some synaptic and nuclear transcription factors) targets.

12: In line 185, it becomes apparent for the first time that two (PRISM) colours are used in parallel. This should also be mentioned in the results/methods section.

A: We have added this important point in the methods section (lines 273-275).

13: Potential miss-classifications of cells are not discussed. Please comment on the accuracy of the classification.

A: We have elaborated on this potential issue in the discussion (lines 220-225). Briefly, while mis-classification is certainly possible, we use multiple markers and have set the classification bar such that a cell must be positive for all designated markers in order to make it into the final analysis group. This allows for automated identification and removal of cells that are not fully fitting into our defined cell identities.

14: The abstract suggests that the presented technique is applied to 3D spheroids and organoid

sections and the discussion confirms that "3D organoid sections (data not shown)" (line 159) have been analysed. However only data from (dissociated) 2D cultures are presented. The claim that 3D samples can be analyzed should be either supported by data or clearly marked as future perspective.

A: We have edited the abstract and manuscript text that refer to 3D spheroids and organoid sections as future perspective.

15: line 297: "assuming that's what you used". Please check.

A: We have removed this note from the revision, and we deeply thank the reviewer for pointing out this embarrassing slip, so that we could correct it.

REVIEWERS' COMMENTS:

Reviewer #2 (Remarks to the Author):

The revised manuscript has been improved based on changes made in response to reviewers' suggestions. In particular, there is a more extensive discussion of the LNA/DNA distinction. References to organoids and patient-specific iPSC data that were not previously shown have also been eliminated. Overall, there is improved clarity throughout. However, I think that a brief mention of the differences between the 2D and 3D derived cultures would be useful. Both 2D and 3D-derived cultures are ultimately cultured in 2D so there's not much of a reason to expect differences between the two with respect to the efficacy of PRISM. However 3D-derived cultures might be more heterogeneous, with cells in much different states, which is an advantage for an approach to integrating multiple markers. Mentioning this at the outset could help the reader understand why the two different culture types are examined and mentioned through the manuscript.

Reviewer #3 (Remarks to the Author):

The authors have addressed most of the concerns and improved the manuscript substantially. The result section is now easier to read and the details of the data acquisition and analysis are clarified.

Besides the following remaining issue, the manuscript is in a good shape for publication:

In comment 9, I raised questions about the listed DNA-docker and imager sequences in SI Table 7. Now, after the authors have clarified the directions, it is apparent that the stated imager sequences cannot bind the docker sequences, as they are complementary. To enable binding they would need to be reverse-complementary. As the microscopy images are convincing, I encourage the authors to double check the stated sequences.

Minor:

line 824: ATOO647N -> ATTO647N

We want to thank the reviewers for their time in reviewing our manuscript and have addressed their comments below. The relevant areas in the manuscript text that were updated are marked in yellow and underlined.

Reviewer 2:

A brief discussion about the differences in 2D vs 3D would also be beneficial.

Clarification on the transition from 3D culture to 2D culture to generate the cortical stains is now provided in the text (lines 34-36; 67; 118-119; 122; 147). We clarify that the neurons are differentiated in 3D suspension culture, and then dissociated into 2D cultures for PRISM staining.

Reviewer 3:

Please check the imager sequences and docker sequences provided.

We have corrected the typographical error in the SI Table 7 (Lines 783-787). Briefly, the docker sequences are in the 5'-3' direction, while the imager sequences are in the 3'-5' direction as shown in the table.